# PrEP awareness and use among reproductive age women in Miami, Florida

Nicholas Fonseca Nogueira[1], Nicole Luisi[2], Ana S. Salazar[1,3], Emily M. Cherenack[4], Patricia Raccamarich[1], Nichole R. Klatt[5], Deborah L. Jones[6], Maria L. Alcaide[1]*

**1** Division of Infectious Diseases, Department of Medicine, University of Miami Miller School of Medicine, Miami, FL, United States of America, **2** Department of Epidemiology, Rollins School of Public Health, Emory University, Atlanta, GA, United States of America, **3** Jackson Memorial Hospital/University of Miami Miller School of Medicine, Miami, FL, United States of America, **4** Department of Public Health Sciences, University of Miami Miller School of Medicine, Miami, FL, United States of America, **5** Division of Surgical Outcomes and Precision Medicine Research, Department of Surgery, University of Minnesota, Minneapolis, MN, United States of America, **6** Department of Psychiatry and Behavioral Sciences, University of Miami Miller School of Medicine, Miami, FL, United States of America

* malcaide@med.miami.edu

## Abstract

### Background

Miami, Florida is an epicenter of the HIV epidemic in the US, with 20% of new HIV infections occurring in women. Despite effectiveness of Pre-Exposure Prophylaxis (PrEP) in preventing HIV, only 10% of eligible women benefit from its use.

### Setting

This study evaluates PrEP awareness and use, and factors associated with PrEP awareness among sexually active women in Miami, Florida.

### Methods

Results reported in this study included cross-sectional data that were collected as part of a baseline visit from a parent study. Cis-gender, HIV-negative, 18-45-year-old, sexually active women were recruited as part of a study evaluating recurrent bacterial vaginosis and HIV risk. Participants completed questionnaires assessing socio-demographics, HIV risk factors, prior history of HIV testing and reproductive tract infections, PrEP awareness and use. Relationships between variables and PrEP awareness were analyzed and multivariable logistic regression identified variables strongly associated with PrEP awareness.

### Results

Among the 295 women enrolled, median age was 31 (24–38) years, 49% Black, 39% White, and 34% Hispanic. Of 63% who knew about PrEP, only 5% were on PrEP. Women with income below poverty line ($OR = 2.00[1.04,3.87]; p = 0.04$), more male sexual partners in past month ($OR = 1.30[1.01,1.68]; p = 0.04$), lifetime HIV testing ($OR = 6.42[2.83,14.52]; p<0.01$), and current bacterial vaginosis ($OR = 2.28[1.18,4.40]; p = 0.01$) were more likely to be aware of PrEP. Lower odds of PrEP awareness were associated with being Black ($OR =$

**Data Availability Statement:** The data for this study include potentially sensitive information related to infectious diseases and sexual behaviors from participants in a limited geographical region.

As the study includes information on HIV, pregnancy history, substance use, and sexual behavior, extra caution must be taken to ensure individuals who access these data have appropriate ethical approvals and data security standards in place. As such, data are available upon reasonable request to study data manager Nicholas Fonseca Nogueira (n.fonsecanogueira@umiami.edu), Principal Investigator Dr. Maria Alcaide (MAlcaide@med.miami.edu), or study biostatistician Dr. Yue Pan (panyue@med.miami.edu).

**Funding:** The data obtained for this project was acquired as part of an on-going longitudinal Women, HIV, Immunology, Microbiome, and Sexual Health (WHIMS) study (R01AI138718 to M.L.A.). This work was supported by National Institutes of Health's grants to the Center for HIV and Research in Mental Health (P30MH116867 to D.L.J.) and the Center for AIDS Research (P30A1073961 to M.L.A.) at the University of Miami. E.M.C's work was supported by the National Institute of Allergy and Infectious Diseases (F32AI162229 to E.M.C). The funders had no role in study design, data collection and analysis, decision to publish, or preparation of the manuscript.

**Competing interests:** The authors have declared that no competing interests exist.

0.38[0.15,0.96];$p = 0.04$), Hispanic ($OR = 0.18[0.08,0.39]$;$p<0.01$), heterosexual ($OR = 0.29$ [0.11,0.77];$p<0.01$), and reporting inconsistent condom use during vaginal sex ($OR = 0.21$ [0.08,0.56];$p<0.01$).

## Conclusion

PrEP awareness is low among reproductive age women in a high-risk setting. Culturally tailored interventions are needed to increase PrEP awareness and uptake, especially among Black and Hispanic women with inconsistent condom use during vaginal sex with male partners.

## Introduction

Despite having only 38% of the country's population, the Southern U.S accounts for 52% of new HIV diagnoses, with about 17.6 persons diagnosed per 100,000 [1, 2]. Florida has one of the highest adult and adolescent incidence and prevalence of HIV in the nation, with 23.7 persons newly diagnosed and 615.0 persons living with HIV per 100,000 persons in 2019 [3, 4]. Approximately 25% of all new HIV cases in Florida are diagnosed in the county of Miami-Dade, with about 20% of new incidences occurring in women [3–6]. Risk factors identified for HIV infection among women include recent sexually transmitted infections (STI), bacterial vaginosis (BV), number of sexual partners, inconsistent condom use, substance use, and partner violence [7, 8]. In addition, heterosexual contact is the primary avenue of new HIV infections across all racial groups; this is more prominent in women identifying as Asian (93%), Black/African American (91%), or Hispanic/Latino (87%) [3]. Factors such as partner violence and substance use may decrease women's ability to use condoms to prevent HIV [9, 10]. Women may also decide not to use condoms because they hope to conceive. This warrants an approach to HIV prevention tailored for the unique needs and experiences of women in the South that is controlled by women, does not rely on condom use, and can be used during the process of conception.

The use of pre-exposure prophylaxis (PrEP) is an effective tool for HIV prevention for eligible individuals when adherence is high [11]. If taken as prescribed, oral pill or long-acting injectable PrEP reduces the risk for sexual acquisition of HIV infection by approximately 99% [12]. Among the estimated number of heterosexually active adults indicated for PrEP use in 2015, approximately 70% were females [13]. Overall, despite the potential effectiveness of PrEP for decreasing HIV acquisition risk, uptake of PrEP among women most at risk for acquiring HIV is low. PrEP-to-need ratio (PnR) for women, defined as the number of PrEP users by new HIV diagnoses, is approximately one-fifth the PnR of men, indicating unmet HIV prevention needs among women [14, 15]. Women account only for 4.7% of filled PrEP prescriptions in 2016 [7, 16–18]. It has been suggested that uptake challenges may stem from knowledge deficits instead of attitudes toward prevention [19], as PrEP awareness is especially low in women, with awareness levels ranging from 10–20% [7, 16].

As PrEP indications are limited to those engaging in sexual practices, recent bacterial STIs, sex partners of unknown/HIV positive status, and inconsistent condom use; additional factors like gender identity, sexual orientation, substance use, and pregnancy intentions may also be important in assessing HIV risk and recommendations for testing and prevention [20, 21]. With many PrEP trials and advertising focused on HIV prevention for men who have sex with men and the LGBTIA+ community, cis-gender heterosexual women are often left out of

research and education efforts, and culturally adapted interventions to increase PrEP aware-ness and uptake among women are growing but still scarce [22]. Given the high percentage of Hispanic women living in Miami, it is necessary to ensure Hispanic women are included in these efforts. As a result, the current study sought to identify factors associated PrEP aware-ness and uptake among sexually active women of reproductive age in Miami-Dade County, Florida.

## Materials and methods

### Recruitment

Recruitment occurred between November 2018 and January 2022 in Miami-Dade County, Florida. Individuals were reached through passive (e.g., flyers, friend & family referrals, and word-of-mouth) and active (e.g., coordinator outreach and community events) efforts as part of an on-going longitudinal study "Women, HIV, Immunology, Microbiome, and Sexual Health" (WHIMS). This study evaluates factors associated with BV recurrence using a multi-disciplinary approach [23]. Referrals from the Florida Department of Health, Center for AIDS Research (CFAR), and the Center for HIV Research and Mental Health (CHARM) contrib-uted significantly to recruitment endeavors. Results reported in this study included cross-sec-tional data that were collected as part of a baseline visit from the parent study. Cis-gender women, between the ages of 18–45, who indicated they were sexually active within the last three months were eligible. Women were not eligible if they were pregnant; immunosup-pressed; diagnosed with HIV; had an intrauterine device; had a history of cervical surgical treatment or cervical intra-epithelia neoplasia (Grade 2 & 3); had recent antibiotic use or allergy/intolerance to metronidazole; or were diagnosed with chlamydia or gonorrhea within the last 2 months. Potentially eligible participants completed laboratory tests to confirm they were negative for pregnancy and HIV. The University of Miami Institutional Review Board (IRB# 20180758) approved the study, and all participants completed written informed consent prior to initiating study assessments.

### Procedures and measures

Participants were screened via telephone for eligibility. Eligible participants completed a self-administered questionnaire remotely (e.g., electronically or by phone) using REDCap. Socio-demographic questions included age, race, ethnicity, employment, income (operationalized as being above or below the poverty line), and educational attainment. Sexual and reproductive health history included lifetime history of any pregnancy, current hormonal contraception use, and lifetime history of any HIV testing. Sexual behavior questions asked women about current relationship status (legally married, unmarried with partner, unmarried, or other), age of first sexual encounter with a man, number of sexual encounters (vaginal sex, oral sex, or anal sex with any partner) in the past month, consistency of condom use during vaginal inter-course with male partners in the past month (always use condoms, sometimes used condoms, never used condoms), number of male sex partners in the past month, number of new male sex partners in the past month, lifetime history of any sexual encounters with women, and sex-ual orientation (heterosexual/straight, bisexual, lesbian/gay, or other). Women were asked if they ever had sex for drugs, money, or shelter. Lifetime substance use was measured as having ever used marijuana, cocaine, crack, heroin, prescription opiate painkillers, methamphet-amine, hallucinogens, club drugs, or any other illicit or recreational drugs. Alcohol use was operationalized as any alcoholic beverages in the past month, while tobacco use was operatio-nalized as any use of tobacco cigarettes, cigars, pipes, or chewing tobacco in the past month. PrEP related outcomes included PrEP awareness, asked as "Have you ever heard about PrEP

(Truvada)? This is a pill that can be taken to prevent HIV infection" and PrEP uptake, asked as "Are you currently taking PrEP?" After the survey, a gynecological exam at the Infectious Diseases Research Unit at the University of Miami was conducted to determine enrollment eligibility for the larger study, at which point participants were tested for BV. The subset of individuals with BV (n = 140) were also tested for STIs (gonorrhea, chlamydia, and trichomonas).

## Statistical analysis

Descriptive analyses characterized the study population. Group comparisons compared the proportion of PrEP aware and unaware participants across demographic, behavioral, and sexual and reproductive health variables using Chi-square, or Fisher's exact test where appropriate due to cell sizes, or studentized t-test. This cross-sectional study was a secondary data analysis to the larger parent study, therefore pre-specification of an analyses strategy prior to initiation of recruitment was not possible. However, a specified primary analysis strategy consisted of a multivariable unconditional logistic regression performed to control for the effects of multiple variables and to obtain maximum likelihood estimates of factors contributing to PrEP awareness. The initial model covariates included demographics (age, race, ethnicity, sexual orientation, below poverty line, educational attainment), any HIV testing, sexual behaviors (age at first male sexual encounter, number of male sexual partners in past month, number of new male sexual partners in past month, condom use with male partners in past month, and sex for drugs, money, or shelter), substance use (any lifetime illicit substance use, any smoking in past month, any alcohol use in past month), and BV diagnosis. STI was not included, because only the subset of participants with BV received STI testing. In cases where items had significant overlap (e.g., sexual orientation and history of sex with women), only one variable was included in the model. Remaining variables were included in the model and backward elimination using $\alpha = 0.25$ was used to select the best-fitting reduced model. Assumption of linearity between independent variables and log-odds was assessed using Box-Tidwell's test, while multicollinearity was assessed by condition indices $\geq 30$ and variance decomposition proportions $\geq 0.8$ [24]. Influential observations were examined in plots using individual delta-beta measurements and Cook's distance. Goodness of fit was evaluated using Hosmer-Lemeshow Statistic. Adjusted odds ratios (aOR) and 95% confidence intervals (CI) were reported and presented as forest plots. Observations with missing values for the response or explanatory variables were excluded. Statistical significance was determined using $\alpha = 0.05$, and analyses were conducted using SAS 9.4.

## Results

Overall, 295 women were enrolled in the study. The sample was diverse in terms of race, ethnicity, and socioeconomic status (Table 1). About 63% of women had heard about PrEP. Out of the 185 women reporting having heard about PrEP, only 5.4% were actively using it. Many HIV acquisition behaviors were generally higher in the PrEP aware compared to the PrEP unaware group. For instance, PrEP awareness associated with current BV diagnosis (72.4% v. 27.6%; p<0.01), greater mean number of male sexual partners in the past month (1.8 v. 1.3; p = 0.04), history of sex for drugs, money, or shelter (85% v. 15%; p = 0.03), and current STI diagnosis (80.6% v. 19.4%; p = 0.03) (Table 2). PrEP awareness was also higher among participants with a lifetime history of sex with women (77% v. 23%; p<0.01).

Covariates included in the multivariable logistic regression after backwards selection were race, ethnicity, below poverty line status, sexual orientation, current diagnosis of Bacterial Vaginosis, lifetime history of HIV testing, condom use during vaginal sex with men, and number

**Table 1. Sociodemographic, history of pregnancy, and contraceptive use by PrEP awareness status (n = 295).**

|  | PrEP Aware* (*n* = 185) | PrEP Unaware* (*n* = 110) | *p* |
|---|---|---|---|
| **Age–years, median (IQR)** | 32 (25–39) | 29 (22–37) | 0.06 |
| **Race–n (%) [a]** |  |  | **<0.01** |
| Black or African American | 107 (74.3) | 37 (25.7) |  |
| White | 61 (52.6) | 55 (47.4) |  |
| Other | 17 (48.6) | 18 (51.4) |  |
| **Ethnicity—n (%)** |  |  | **<0.01** |
| Hispanic | 43 (43.4) | 56 (56.6) |  |
| Non-Hispanic | 142 (72.4) | 54 (27.6) |  |
| **Heterosexual—n (%)** | 135 (57.5) | 100 (42.6) | **<0.01** |
| **Relationship Status–n (%) [b]** |  |  | 0.59 |
| Legally Married | 19 (55.9) | 15 (44.1) |  |
| Unmarried with Partner | 35 (70.0) | 15 (30.0) |  |
| Unmarried | 107 (62.6) | 64 (37.4) |  |
| Other | 24 (60.0) | 16 (40.0) |  |
| **Employment (Full or Part-time)–n (%)** | 75 (58.1) | 54 (41.9) | 0.15 |
| **Below Poverty Line–n (%)** | 98 (72.6) | 37 (27.4) | **<0.01** |
| **Educational Attainment–n (%)** |  |  | 0.32 |
| Less than High School | 31 (75.6) | 10 (24.4) |  |
| Completed High School | 56 (62.2) | 34 (37.8) |  |
| Some College or Associates Degree | 49 (57.7) | 36 (42.4) |  |
| Completed 4 years of college | 31 (58.5) | 22 (41.5) |  |
| Attended or Completed Graduate School | 18 (69.2) | 8 (30.8) |  |
| **Lifetime History of Any Pregnancy–n (%)** | 112 (67.5) | 54 (32.5) | 0.06 |
| **Contraception Use[c] –n (%)** | 27 (52.9) | 24 (47.1) | 0.11 |

Notes.

*PrEP Aware—persons who heard about PrEP; PrEP Unaware–persons who never heard about PrEP.

[a]Asian, Native American, Native Hawaiian or pacific islander were collapsed into Other.

[b]Widowed, divorced, separated, and never married were collapsed into Unmarried.

[c]Contraceptive use refers to pills, patch, ring, implant (etonosgestrel), or depot (DMPA).

of male sexual partners in the past month. A total of 247 observations were evaluated to obtain maximum likelihood estimates in the model. Observations with missing values (n = 48 observations) for the response or explanatory variables were excluded. A non-significant Box-Tidwell test indicated no violations of linearity assumption. There was no evidence of multicollinearity in the selected model (largest CNI = 12.3). Hosmer and Lemeshow goodness-of-fit test indicated no evidence of lack of fit ($X^2$ = 6.9; df = 8; *p* = 0.55). Adjusted Odds Ratios are presented in Fig 1. In multivariable logistic regression, women with average household income below poverty line (OR = 2.00 [1.04,3.87]; p = 0.04), more male sexual partners in the past month (OR = 1.30 [1.01,1.68]; p = 0.04), any prior HIV testing (OR = 6.42 [2.83,14.52]; p<0.01), and current BV (OR = 2.28 [1.18,4.40]; p = 0.01) were more likely to be aware of PrEP. Lower odds of PrEP awareness were associated with being Black compared to white (OR = 0.38 [0.15,0.96]; p = 0.04), Hispanic compared to non-Hispanic (OR = 0.18 [0.08,0.39]; p<0.01), identifying as heterosexual compared to bisexual, lesbian/gay, or other (OR = 0.29 [0.11,0.77]; p<0.01), and reporting inconsistent condom use compared to always using condoms with male partners in the past month (OR = 0.21 [0.08,0.56]; p<0.01).

**Table 2. Risk behaviors, BV, and STI by PrEP awareness status (n = 295).**

| | PrEP Aware* (n = 185) | PrEP Unaware* (n = 110) | p |
|---|---|---|---|
| **Current PrEP use–n (%)** | 10 (100.0) | N/A | N/A |
| **Lifetime History of HIV Test–n (%)** | 159 (70.7) | 66 (29.3) | <**0.01** |
| **Current Diagnosis of Bacterial Vaginosis–n (%)** | 105 (72.4) | 40 (27.6) | <**0.01** |
| Lifetime History of Any Substance Use [a] –n (%) | 98 (65.3) | 52 (34.7) | 0.34 |
| **Any Alcohol Use in Past Month–n (%)** | 109 (63.4) | 63 (36.6) | 0.78 |
| **Age at First Sexual Encounter–median (IQR)** | 16 (14–18) | 17 (15–18) | 0.12 |
| **Any Sexual Encounters in Past Month [b] –(%)** | 161 (62.4) | 97 (37.6) | 0.77 |
| **Condom Use during Vaginal Sex in Past Month–n (%)** | | | 0.11 |
| Always used Condoms | 41 (69.5) | 18 (30.5) | |
| Sometimes used Condoms | 25 (50.0) | 25 (50.0) | |
| Never used Condoms | 88 (62.9) | 52 (37.1) | |
| **New Male Sexual Partners in Past Month–n (%)** | 34 (54.0) | 29 (46.0) | 0.11 |
| **Number of Male Sexual Partners in Past Month–μ ± SD** | 1.8 ± 3.15 | 1.3 ± 0.95 | **0.04** |
| **Number of Male Sexual Partners in past 5 years—μ ± SD** | 5.5 ± 7.95 | 4.5 ± 5.98 | 0.25 |
| **Number of Sexual Encounters in Past Month [b] - μ ± SD** | 6.5 ± 7.92 | 6.8 ± 7.33 | 0.68 |
| **Lifetime History of Sex with Women–n (%)** | 57 (77.0) | 17 (23.0) | <**0.01** |
| **Lifetime History of Sex for Drugs, Money, or Shelter–n (%)** | 17 (85.0) | 3 (15.0) | **0.03** |
| **Current STI [c] (n = 140)—n (%)** | 29 (80.6) | 7 (19.4) | 0.19 |

Notes.

*PrEP Aware—persons who heard about PrEP; PrEP Unware–persons who never heard about PrEP.

[a]Substance use includes medical or recreational marijuana, cocaine, crack, heroin, methamphetamine, hallucinogens, club drugs, or any other illicit or recreational drugs.

[b]Sexual Encounters includes any vaginal, oral, or anal sex with any partner.

[c]Sexually transmitted infections were laboratory confirmed and include gonorrhea, chlamydia, and trichomonas.

## Discussion

Miami, a culturally and ethnically diverse city, has one of the highest HIV incidences and prevalence in the United States [3, 5, 6]. This study assessed factors associated with PrEP awareness in sexually active women of reproductive age in Miami-Dade County, Florida. Results show that although most women had heard about PrEP (62.7%), there was a low reported PrEP use among this group (5.4%). This is consistent with the national PrEP use previously observed in the literature [7, 16, 17]. This study is novel as it highlighted the unique needs and experiences of women in the Southern U.S. with a specific focus around Hispanic women.

Women with a lifetime history of HIV testing had over six times the odds of PrEP awareness. This could indicate that women who perceive themselves as being at risk for acquiring HIV are more likely to seek education about both PrEP and HIV testing as options to reduce risk. It could also indicate that providers are using HIV testing visits as an opportunity to provide PrEP education to women. Research on the timing of PrEP education supports the notion that there are opportunities to provide education at specific timepoints, such as STI screenings, peri-conception, and contraceptive counseling [8, 25, 26]. Although women diagnosed with BV at the study visit were more likely to have heard about PrEP when controlling for HIV testing and sexual behaviors, it is unknown if the relationship of BV to PrEP awareness is at least partially mediated by increased levels of perceived HIV risk, increased contact with healthcare providers due to reoccurring BV, or another reason. It was surprising to note that the women

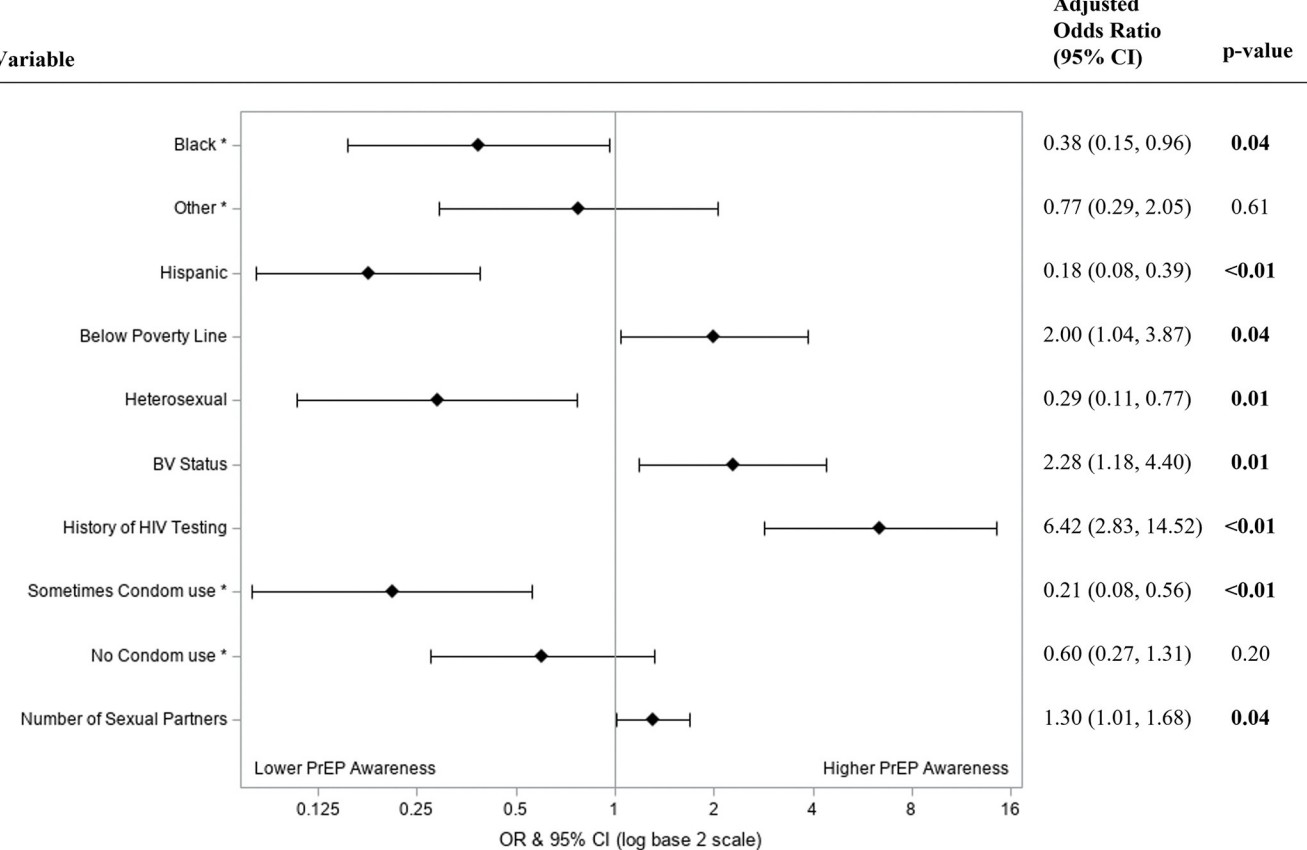

| Variable | Adjusted Odds Ratio (95% CI) | p-value |
|---|---|---|
| Black * | 0.38 (0.15, 0.96) | **0.04** |
| Other * | 0.77 (0.29, 2.05) | 0.61 |
| Hispanic | 0.18 (0.08, 0.39) | **<0.01** |
| Below Poverty Line | 2.00 (1.04, 3.87) | **0.04** |
| Heterosexual | 0.29 (0.11, 0.77) | **0.01** |
| BV Status | 2.28 (1.18, 4.40) | **0.01** |
| History of HIV Testing | 6.42 (2.83, 14.52) | **<0.01** |
| Sometimes Condom use * | 0.21 (0.08, 0.56) | **<0.01** |
| No Condom use * | 0.60 (0.27, 1.31) | 0.20 |
| Number of Sexual Partners | 1.30 (1.01, 1.68) | **0.04** |

*Note. Reference Category for Race was White & Condom Use was "Always"

**Fig 1. Adjusted odds ratios for PrEP awareness, by factors (n = 247).** Factors associated with PrEP awareness in a multivariable logistic regression model containing race, ethnicity, poverty line status, sexual orientation, current diagnosis of bacterial vaginosis, lifetime history of HIV testing, past-month condom uses with men, and number of male sexual partners. Adjusted odd ratios (aORs) were calculated and presented using a log base 2 scale in a Forest Plot. Null line is indicated for no covariate effects and bolded lines represent aORs with 95% confidence intervals. Bolded lines above and below the null line indicate increased or decreased odds of PrEP awareness, respectively.

below the poverty line were more likely to be aware of PrEP. Women living below the poverty line may more heavily depend on community health centers or public health clinics for their healthcare needs where they have a higher chance of encountering information surrounding PrEP or HIV prevention education. Alternatively, community outreach efforts or public health campaigns may specifically target economically disadvantaged groups who have limited access to healthcare/ resources (e.g., condoms) or engage in risky behaviors that increase susceptibility to HIV. However, education alone will not be sufficient to address PrEP uptake and other financial barriers to PrEP.

Our findings are in agreement with other reports highlighting ethnic and racial disparities in PrEP awareness and use [16]. Reports indicate that while Hispanic women welcome the idea of using PrEP [27], PrEP awareness is significantly lower than in non-Hispanics, especially among Hispanic adolescents aged 14–17 [28]. Given the low awareness and high willingness to use PrEP among Hispanic women [29], culturally appropriate, sexual education initiatives present a clear and effective opportunity in the delivery of PrEP information and services in ethnically diverse areas. Similarly to Bush et al. [30], we found that Black women were less likely to have heard about PrEP than White women. This suggests that additional efforts are needed that are tailored to these groups.

Findings that heterosexual women and women with inconsistent condom use during intercourse are less likely to know about PrEP are concerning, because women who have sex with men are those most at risk for acquiring HIV, while the risk for HIV acquisition for women who have sex exclusively with women lower [31]. It is possible that women who access LGBTQIA+ community resources are exposed to the higher levels of PrEP messaging geared towards sexual minorities, and additional messaging is needed in community locations frequented by heterosexual women.

This study is limited by its cross-sectional design and self-reported data; thus, results cannot be interpreted as causal. Also, it is limited in that we were not powered to examine factors associated with PrEP uptake due to low uptake rates. We were also unable to determine whether women had medical contraindications for PrEP, which could limit providers' willingness to provide PrEP education. Given this sample represents an ethnically diverse population in Miami, its generalizability may be limited to less diverse areas across the country. Furthermore, as of 2022, PrEP is now available in Florida at no cost to individuals. As a result, individuals residing in states with more restricting PrEP coverage policies may be subjected to unique sociopolitical environments that differ when compared to Miami. In addition, PrEP awareness may have been influenced by broad media campaigns from the pharmaceutical industry that have occurred in recent years. Larger studies are needed to better understand health behaviors for willingness to use PrEP, which could guide interventions to encourage and increase PrEP uptake and adherence. Effectively increasing PrEP awareness and ultimately, uptake, requires a multifaceted approach targeting various barriers such as reducing barriers to PrEP initiation/maintenance, improving risk perceptions, enhancing insurance coverage and Medicaid expansion in Florida, reducing stigma, and remodeling structural characteristics of healthcare services [11].

## Conclusions

Our findings suggest that women with lower incomes, more male sexual partners, a prior history of HIV testing, and current bacterial vaginosis were more likely to be aware of PrEP, while lower odds of PrEP awareness were associated with being Black, Hispanic, heterosexual, and reporting inconsistent condom use with male partners. Overall, our study suggests that PrEP education efforts in Miami-Dade County have been moderately successful in reaching women, but there is still a need for more focused outreach towards Black, Hispanic, and heterosexual women as notable disparities in PrEP awareness among these demographic groups persist. These results suggest that targeted interventions are needed to address the specific needs and challenges of these demographic groups. In prior research, obstacles to PrEP initiation among women have included culturally appropriate framing of PrEP information, concerns about effects of long-term medication use, social cues, insurance coverage, and challenges in identifying women with an increased HIV-risk [7, 18, 32]. Studies have suggested improved socioeconomical resources, PrEP knowledge, health literary, healthcare accessibility, insurance coverage, and heightening perceived HIV risk as potential targets for increasing PrEP initiation [11, 18, 33]. In summary, our results highlight the need for multi-level interventions to increase PrEP awareness and uptake among Black, Hispanic, and heterosexual women in South Florida.

## Supporting information

**S1 Table. Multivariable logistic regression assessing factors associated with PrEP awareness (n = 247)[a].**
(DOCX)

## Author Contributions

**Conceptualization:** Nicholas Fonseca Nogueira, Nichole R. Klatt, Maria L. Alcaide.

**Data curation:** Nicholas Fonseca Nogueira, Nicole Luisi.

**Formal analysis:** Nicholas Fonseca Nogueira, Nicole Luisi.

**Funding acquisition:** Nichole R. Klatt, Maria L. Alcaide.

**Investigation:** Nicholas Fonseca Nogueira, Nicole Luisi, Ana S. Salazar, Emily M. Cherenack, Patricia Raccamarich, Nichole R. Klatt, Deborah L. Jones, Maria L. Alcaide.

**Methodology:** Nicholas Fonseca Nogueira, Nicole Luisi.

**Project administration:** Nicholas Fonseca Nogueira, Nichole R. Klatt, Maria L. Alcaide.

**Resources:** Maria L. Alcaide.

**Supervision:** Patricia Raccamarich, Maria L. Alcaide.

**Validation:** Nicholas Fonseca Nogueira, Nicole Luisi.

**Visualization:** Nicholas Fonseca Nogueira, Nicole Luisi.

**Writing – original draft:** Nicholas Fonseca Nogueira, Nicole Luisi, Ana S. Salazar, Emily M. Cherenack, Patricia Raccamarich, Nichole R. Klatt, Deborah L. Jones, Maria L. Alcaide.

**Writing – review & editing:** Nicholas Fonseca Nogueira, Nicole Luisi, Ana S. Salazar, Emily M. Cherenack, Patricia Raccamarich, Nichole R. Klatt, Deborah L. Jones, Maria L. Alcaide.

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
