## [Decision Letter · Decision Letter 0]

28 Feb 2023

PONE-D-23-01029PrEP awareness and use among reproductive age women in Miami, Florida.PLOS ONE

Dear Dr. Alcaide,

Thank you for submitting your manuscript to PLOS ONE. After careful consideration, we feel that it has merit but does not fully meet PLOS ONE’s publication criteria as it currently stands. Therefore, we invite you to submit a revised version of the manuscript that addresses the points raised during the review process.

We look forward to receiving your revised manuscript.

Kind regards,

Fengyi Jin, Ph.D.

Academic Editor

PLOS ONE

Journal Requirements:

Additional Editor Comments:

Thank you for your submission to PLOS ONE. Your manuscript has now been reviewed by two experts in the field and some comments are raised during the process.

In your revision, please pay close attention to the comments raised and address them in your point-to-point response for easier cross-referencing.

Reviewers' comments:

Reviewer's Responses to Questions

**Comments to the Author**

1. Is the manuscript technically sound, and do the data support the conclusions?

Reviewer #1: Yes

Reviewer #2: Partly

2. Has the statistical analysis been performed appropriately and rigorously? 

Reviewer #1: Yes

Reviewer #2: Yes

3. Have the authors made all data underlying the findings in their manuscript fully available?

Reviewer #1: No

Reviewer #2: No

4. Is the manuscript presented in an intelligible fashion and written in standard English?

Reviewer #1: Yes

Reviewer #2: Yes

5. Review Comments to the Author

Reviewer #1: The manuscripts describes findings from a study that evaluated PrEP awareness and use, and factors associated with PrEP awareness among sexually active women in Miami, Florida. The findings significantly contribute to the field of HIV prevalence and reducing health disparities among minority women, at risk of HIV. The paper is very well written. The findings are sound and the discussion is fully supported by study findings. The authors did an excellent job.

No suggested revisions.

Reviewer #2: Thank you for the opportunity to review this interesting manuscript about PrEP awareness and associated factors among women in Florida. This manuscript shows important results for public health consideration in increasing PrEP awareness among specific groups of women. I have listed below my comments to help improve the quality of this manuscript.

1. Methods.

Could the authors be more explicit in describing the study design?

I assume the data shown are from cross-sectional analyses of a longitudinal study? It should be clearly explained if the data used were the baseline data (recruitment stage). Please also mention this in the abstract.

2. Methods.

It is noted that awareness was >60%, but the (current) uptake was low in this study.

However, did the authors also consider assessing PrEP use willingness or the use of PrEP in the past? (if data available).

The readers would benefit more from this information. It is important to also know the proportion who did not take PrEP but actually intended to use it, and what the barriers of not using PrEP are.

3. Methods.

statistical analysis: it would be useful to add whether the statistical analysis was pre-specified and, if so, provide a statistical analysis plan.

For association with age, consider to make it a categorical variable, e.g younger vs.older; broader category of age.

4. Results.

It said that 295 women were enrolled in the study. Were data from all 295 women used? Were there any missing data?

Please clarify and explain this somewhere in the results. Also how did authors treat the missing data (if any).

5. Results.

Please be careful with using the word 'predictor' (line 201). As this is a cross-sectional analysis, it is difficult to interpret the temporality.

6. Discussion.

I do not follow the statement that it is encouraging the women below the poverty line were likely to be aware of PrEP (lines 229...). How it is encouraging? Please elaborate more on the association between PrEP awareness and economic disadvantage.

7. Discussion.

Also I'd like to see a much more thorough discussion on the feasibility of scaling up PrEP uptake among eligible women. Making PrEP available freely is clearly not sufficient to achieve high uptake among women. It's not clear whether it would be easy or difficult to implement such interventions to increase PrEP uptake in Miami, or what resources are required.

8. Discussion.

Limitations: What about the limitations of the study being cross-sectional and that independent variables relied on self-report? Self-report data might be influenced by social expectancies and recall biases, particularly among women.

9. Conclusions

I agree with the statements in general, but would appreciate more clarity on how the results of this study lead to the conclusion.

6. PLOS authors have the option to publish the peer review history of their article (what does this mean?). If published, this will include your full peer review and any attached files.

Reviewer #1: No

Reviewer #2: No

---

## [Author Response · Author response to Decision Letter 0]

7 Apr 2023

Reviewer’s Comments : 

1. If there are ethical or legal restrictions on sharing a de-identified data set, please explain them in detail (e.g., data contain potentially sensitive information, data are owned by a third-party organization, etc.) and who has imposed them (e.g., an ethics committee). Please also provide contact information for a data access committee, ethics committee, or other institutional body to which data requests may be sent.

Response: The data for this study include potentially sensitive information related to infectious diseases and sexual behaviors from participants in a limited geographical region. As the study includes information on HIV, pregnancy history, substance use, and sexual behavior, extra caution must be taken to ensure individuals who access these data have appropriate ethical approvals and data security standards in place. As such, data are available upon reasonable request to study data manager Nicholas Fonseca Nogueira (n.fonsecanogueira@umiami.edu) and Principal Investigator Dr. Maria Alcaide (MAlcaide@med.miami.edu).

Reviewer #2: Thank you for the opportunity to review this interesting manuscript about PrEP awareness and associated factors among women in Florida. This manuscript shows important results for public health consideration in increasing PrEP awareness among specific groups of women. I have listed below my comments to help improve the quality of this manuscript.

1. Methods. Could the authors be more explicit in describing the study design? I assume the data shown are from cross-sectional analyses of a longitudinal study? It should be clearly explained if the data used were the baseline data (recruitment stage). Please also mention this in the abstract.

Response: We have added a better description of the study design in our methods and abstract sections (lines 112): “Results reported in this study included cross-sectional data that were collected as part of a baseline visit from the parent study”. 

2. Methods. It is noted that awareness was >60%, but the (current) uptake was low in this study. However, did the authors also consider assessing PrEP use willingness or the use of PrEP in the past? (if data available). The readers would benefit more from this information. It is important to also know the proportion who did not take PrEP but actually intended to use it, and what the barriers of not using PrEP are.

Response: We agree with the reviewer’s comment that this variable would be interesting to

investigate. However, the parent study did not include these variables in the baseline questionnaire, and therefore data is not available. We have included this information in the limitations section of our discussion (lines 277): “Larger studies are needed to better understand health behaviors for willingness to use PrEP, which could guide interventions to encourage and increase PrEP uptake and adherence.” 

3. Methods. statistical analysis: it would be useful to add whether the statistical analysis was pre-specified and, if so, provide a statistical analysis plan. For association with age, consider to make it a categorical variable, e.g younger vs. older; broader category of age.

Response: We understand the reviewer concern for bias and have further clarified the extent of the pre-specified statistical analysis in the Methods. The analysis population is described as (lines 113) “Cis-gender women, between the ages of 18-45, who indicated they were sexually active within the last three months were eligible.” Further descriptions of the statistical modeling, covariate selection, and handling of missingness are outlined in the statistical analysis subsection of the Methods (lines 155): “This cross-sectional study was a secondary data analysis to the larger parent study, therefore pre-specification of an analyses strategy prior to initiation of recruitment was not possible. However, a specified primary analysis strategy consisted of a multivariable unconditional logistic regression performed to control for the effects of multiple variables and to obtain maximum likelihood estimates of factors contributing to PrEP awareness …” We felt like treating age as continuous was more appropriate to provide insights into PrEP awareness in this population and in generating a more parsimonious model. In addition, there were no linearity assumption violation with age or any of the other independent variables included. 

4. Results. It said that 295 women were enrolled in the study. Were data from all 295 women used? Were there any missing data? Please clarify and explain this somewhere in the results. Also how did authors treat the missing data (if any).

Response: Although 295 women were included in the descriptive statistics of the project, we have further revised how missing data was handled surrounding the logistic regression model in the results for clarity (lines 195): “A total of 247 observations were evaluated to obtain maximum likelihood estimates in the model. Observations with missing values (n=48 observations) for the response or explanatory variables were excluded.”

5. Results. Please be careful with using the word 'predictor' (line 201). As this is a cross-sectional analysis, it is difficult to interpret the temporality.

Response: We agree with the reviewer that predictions cannot be assessed by cross-sectional studies. We have included a statement in the discussion section alluding to the above (lines 267): “This study is limited by its cross-sectional design and self-reported data; thus, results cannot be interpreted as causal”. In addition, we have revised language from “predictors of” throughout the manuscript to “factors associated with”. 

6. Discussion. I do not follow the statement that it is encouraging the women below the poverty line were likely to be aware of PrEP (lines 229...). How it is encouraging? Please elaborate more on the association between PrEP awareness and economic disadvantage.

Response: We have elaborated on this in the discussion section (lines 239): “It was surprising to note that the women below the poverty line were more likely to be aware of PrEP. Women living below the poverty line may more heavily depend on community health centers or public health clinics for their healthcare needs where they have a higher chance of encountering information surrounding PrEP or HIV prevention education. Alternatively, community outreach efforts or public health campaigns may specifically target economically disadvantaged groups who have limited access to healthcare/ resources (e.g., condoms) or engage in risky behaviors that increase susceptibility to HIV. However, education alone will not be sufficient to address PrEP uptake and other financial barriers to PrEP.”

7. Discussion. Also I'd like to see a much more thorough discussion on the feasibility of scaling up PrEP uptake among eligible women. Making PrEP available freely is clearly not sufficient to achieve high uptake among women. It's not clear whether it would be easy or difficult to implement such interventions to increase PrEP uptake in Miami, or what resources are required.

Response: We agree with the reviewer that improving PrEP uptake is a complex issue. Although this study was not powered to examine factors associated with PrEP uptake due to low uptake rates, we have expanded on this topic in the discussion (lines 279): “Effectively increasing PrEP awareness and ultimately, uptake, requires a multifaceted approach targeting various barriers such as reducing barriers to PrEP initiation/maintenance, improving risk perceptions, enhancing insurance coverage and Medicaid expansion in Florida, reducing stigma, and remodeling structural characteristics of healthcare services” 

8. Discussion/Limitations: What about the limitations of the study being cross-sectional and that independent variables relied on self-report? Self-report data might be influenced by social expectancies and recall biases, particularly among women.

Response: We agree with this comment. We have included our study design (cross-sectional) and self-report data among the limitations of our study (lines 267): “This study is limited by its cross-sectional design and self-reported data; thus, results cannot be interpreted as causal”

9. Conclusions: I agree with the statements in general but would appreciate more clarity on how the results of this study lead to the conclusion.

Response: We have revised the conclusion statement to link back to the results (lines 285): “Our findings indicate that women with lower incomes, more male sexual partners, a prior history of HIV testing, and current bacterial vaginosis were more likely to be aware of PrEP, while lower odds of PrEP awareness were associated with being Black, Hispanic, heterosexual, and reporting inconsistent condom use with male partners. Overall, our study suggests that PrEP education efforts in Miami-Dade County have been moderately successful in reaching women, but there is still a need for more focused outreach towards Black, Hispanic, and heterosexual women as notable disparities in PrEP awareness among these demographic groups persist. These results suggest that targeted interventions are needed to address the specific needs and challenges of these demographic groups … In summary, our results highlight the need for multi-level interventions to increase PrEP awareness and uptake among Black, Hispanic, and heterosexual women in South Florida.”

---

## [Decision Letter · Decision Letter 1]

9 May 2023

PrEP awareness and use among reproductive age women in Miami, Florida.

PONE-D-23-01029R1

Dear Dr. Alcaide,

We’re pleased to inform you that your manuscript has been judged scientifically suitable for publication and will be formally accepted for publication once it meets all outstanding technical requirements.

Kind regards,

Fengyi Jin, Ph.D.

Academic Editor

PLOS ONE

Additional Editor Comments (optional):

Reviewers' comments:

Reviewer's Responses to Questions

**Comments to the Author**

1. If the authors have adequately addressed your comments raised in a previous round of review and you feel that this manuscript is now acceptable for publication, you may indicate that here to bypass the “Comments to the Author” section, enter your conflict of interest statement in the “Confidential to Editor” section, and submit your "Accept" recommendation.

Reviewer #1: All comments have been addressed

2. Is the manuscript technically sound, and do the data support the conclusions?

Reviewer #1: Yes

3. Has the statistical analysis been performed appropriately and rigorously? 

Reviewer #1: Yes

4. Have the authors made all data underlying the findings in their manuscript fully available?

Reviewer #1: Yes

5. Is the manuscript presented in an intelligible fashion and written in standard English?

Reviewer #1: Yes

6. Review Comments to the Author

Reviewer #1: I have no additional comments to the authors. All concerns have been addressed in the revised version.

7. PLOS authors have the option to publish the peer review history of their article (what does this mean?). If published, this will include your full peer review and any attached files.

Reviewer #1: No

---

## [Editor Report · Acceptance letter]

30 May 2023

PONE-D-23-01029R1 

PrEP awareness and use among reproductive age women in Miami, Florida. 

Dear Dr. Alcaide:

I'm pleased to inform you that your manuscript has been deemed suitable for publication in PLOS ONE. Congratulations! Your manuscript is now with our production department. 

Kind regards, 

on behalf of

Dr. Fengyi Jin 

Academic Editor

PLOS ONE